# The Metabolic Potential of Endophytic *Actinobacteria* Associated with Medicinal Plant *Thymus roseus* as a Plant-Growth Stimulator

**DOI:** 10.3390/microorganisms10091802

**Published:** 2022-09-07

**Authors:** Osama Abdalla Abdelshafy Mohamad, Yong-Hong Liu, Yin Huang, Li Li, Jin-Biao Ma, Dilfuza Egamberdieva, Lei Gao, Bao-Zhu Fang, Shaimaa Hatab, Hong-Chen Jiang, Wen-Jun Li

**Affiliations:** 1State Key Laboratory of Desert and Oasis Ecology, Xinjiang Institute of Ecology and Geography, Chinese Academy of Sciences, Urumqi 830011, China; 2Department of Biological, Marine Sciences and Environmental Agriculture, Institute for Post Graduate Environmental Studies, Arish University, Al-Arish 45511, Egypt; 3Department of Environmental Protection, Faculty of Environmental Agricultural Sciences, Arish University, Al-Arish 45511, Egypt; 4Faculty of Biology, National University of Uzbekistan, Tashkent 100174, Uzbekistan; 5Institute of Fundamental and Applied Research, National Research University (TIIAME), Tashkent 100000, Uzbekistan; 6Faculty of Organic Agriculture, Heliopolis University, Cairo 2834, Egypt; 7State Key Laboratory of Biogeology and Environmental Geology, China University of Geosciences, Wuhan 430074, China; 8State Key Laboratory of Biocontrol, Guangdong Provincial Key Laboratory of Plant Resources, School of Life Sciences, Sun Yat-sen University, Guangzhou 510275, China

**Keywords:** agriculture sustainability, environmental microbiology, medicinal plants, endophytes, biofertilizer, biocontrol, *Fusarium oxysporum*, *Verticillium dahliae*, actinobacteria, *Thymus roseus*

## Abstract

Bio-fertilizer practice considers not only economical but also environmentally friendly, sustainable agriculture. Endophytes can play important beneficiary roles in plant development, directly, indirectly, or synergistically. In this study, the majority of our endophytic actinobacteria were able to possess direct plant growth-promoting (PGP) traits, including auxin (88%), ammonia (96%), siderophore production (94%), and phosphate solubilization (24%), along with cell-wall degrading enzymes such as protease (75%), cellulase (81%), lipase (81%), and chitinase (18%). About 45% of tested strains have an inhibitory effect on the phytopathogen *Fusarium oxysporum*, followed by 26% for *Verticillium dahlia*. Overall, our results showed that strains XIEG63 and XIEG55 were the potent strains with various PGP traits that caused a higher significant increase (*p* ≤ 0.05) in length and biomass in the aerial part and roots of tomato and cotton, compared to the uninoculated plants. Our data showed that the greatest inhibition percentages of two phytopathogens were achieved due to treatment with strains XIEG05, XIEG07, XIEG45, and XIEG51. The GC-MS analysis showed that most of the compounds were mainly alkanes, fatty acid esters, phenols, alkenes, and aromatic chemicals and have been reported to have antifungal activity. Our investigation emphasizes that endophytic actinobacteria associated with medicinal plants might help reduce the use of chemical fertilization and potentially lead to increased agricultural productivity and sustainability.

## 1. Introduction

Progressive global climate change in the twenty-first century is increasingly becoming a menace to all life on earth. Humanity has always been concerned about food production to attend to the incredible increases in population and, for a long time, the solution was to expand agriculture to new areas. The Food and Agricultural Organization (FAO) expects an increasing human population to reach approximately 10 billion people living in the world by 2030 [1,2]. In the world, the climatic change and food security predicted and estimated that the agriculture sector will face the challenge of expanding food production by 70% in the year 2050 to meet the increased demand without causing significant price impacts and shortages [3]. The combination of the ever-growing human population, global climate change, and an evident rise in environmentally destructive human activities, such as deforestation and the overuse of chemical fertilizers and pesticides in agriculture, especially in developing countries, leads to the anticipated increase in the demand for environmentally sustainable agriculture. Thus, the need for a step-change advancement in the agriculture sector has been highlighted; where biotechnological progress in the management of crops is necessary not only to ensure sufficient crop production for current and future generations but also to protect both environmental and human health [4]. Based on the current status of available knowledge and technologies, the successful application of long-term and persistent plant beneficial microbial communities will greatly benefit current and future agricultural outcomes. 

The excessive and continuous application of chemical fertilizers has often caused an unfavorable change in soil ecosystems in matters of structure and fertility and also detrimental effects on human health [5,6]. The context of the development of a novel research frontier in sustainable agricultural practices is represented by the alteration of plant-associated microbiomes in situ to improve plant growth, which might help to shrink the use of agrochemicals [7,8]. In this sense, several studies have been focused on that group of individual microorganisms exerting plant growth promotion properties that naturally inhabit a synergistic association with plants and benefit their host by improving their growth performance by several mechanisms [9,10]. 

In plants, bacterial endophytes are ubiquitous in nature and inhabit every internal part of the root, stem, leaf, and seed for their life cycle partly or entirely in a symbiotic relationship without subjecting them to any disadvantage [11,12,13,14]. Recently, new developments in microbial ecology suggest, in some of the research evidence that has been reported, that bacterial endophytes play significant roles in enhancing plant growth and development by accelerating the availability of major mineral nutrients (e.g., N, phosphorus (P), zinc (Zn), and iron (Fe)), helping in the production of phytohormones (e.g., indole-3-acetic acid, siderophores, and activating systemic resistance against phytopathogenic in plants by producing hydrolyzing enzymes such as cellulases and chitinase which cause the degradation of the fungal cell wall or by the lysis of hyphae and limiting spore germination [12,15,16,17]). Indeed, numerous published literature has assessed that the inoculation of bacterial endophytes positively affected plant growth parameters including root length, shoot length, fresh and dry roots, and shoot biomass in comparison with plants cultivated without inoculation with the endophytes of many commercial plants such as maize [18], strawberry [19,20], curcuma [21], cucumber [22], sunflower [23], cereal crops [24], legumes [25], and tea [15]. 

Actinobacteria are well-known producers of a broad spectrum of bioactive secondary molecules such as antibiotics, enzymes, and antioxidants [26,27]. Thus, scientists are now focusing on the isolation of actinobacteria from varied habitats such as extreme environments [28,29] and the inner tissues of medicinal plants in arid lands [30]. Endophytic actinobacteria have been shown significant interest in recent years for their role in improving plant growth through various mechanisms, including nutrient uptake by producing biologically active secondary metabolites that stimulate plant growth directly through the solubilization of nutrients, the modulation of hormone levels, nitrogen fixation, and increasing the resilience of the plants against environmental stress [31,32,33]. 

Over the years, according to the World Health Organization [34], more than 75% of the world’s population, especially in less developed nations, have relied mainly on herbal medicines for treatment. Moreover, it has been well documented that those endophytic microorganisms associated with herbal medicines are promising candidates for plant-growth promotion. However, very few investigations are available on the beneficial endophytic actinobacteria from arid plants, thus there are good opportunities for screening endophytic actinobacteria for their functional roles, which is a promising approach for the development of plant growth-promoting microbes as bio-stimulants to increase crop productivity [10,35,36]. The application of such microbial-based crop amendments is rapidly growing globally [37] and could act as a promising alternative technique to some traditional agricultural techniques, especially in countries where agriculture is the main source of economic development. It is assumed that developing countries in Asia and Africa have the potential to largely benefit from the application of biofertilizers developed from beneficial endophytic bacterial populations associated with medicinal plants, with a predicted increase in crop yields of up to 10% [38]. 

*Thymus roseus* is a traditional medicinal herb in the mint family (*Lamiaceae*) and *Thymus* essential oils are used worldwide in pharmaceutical and food applications [39,40]. Tomato and cotton are economically important crops worldwide; however, they are facing several biotic and abiotic stresses and are sensitive to vascular wilt diseases by *Fusarium oxysporum* and *Verticillium dahlia*, respectively [41,42]. Recently, a novel research frontier in agriculture is represented by the alteration of plant-associated microbiomes in situ to improve plant growth performance. However, our understanding of functions and mechanistic plant microbe’s interactions for these communications remain extremely limited, hampering our ability to fully benefit from these promising tools. On the basis of these premises, the present research aimed to assess the impact of cultivated beneficial endophytic actinobacterial strains associated with wild *T. roseus* in enhancing tomato and cotton crop yields grown in low-nutrient soil mixtures and phytopathogen growth inhibition with the aim of investigating whether the growth promotion properties of the tested strains were restricted to cotton and tomato or not. To test this hypothesis, we previously cultured and identified 55 endophytic actinobacterial strains [43] and will functionally characterize them in vitro and in planta assays. Additionally, we will evaluate their ability to biocontrol the economically important plant phytopathogens *V. dahliae* and *F. oxysporum* in vitro and identify the major antifungal and antimicrobial compounds produced by endophytes.

## 2. Materials and Methods

### 2.1. In Vitro Screening of Potent Bacterial Endophytes for Plant Beneficial Traits

All bacterial endophytes, 55 strains, including 2 classes, 8 orders, 12 families, 18 genera, and 37 species isolated from the Ili site [43], were screened for direct PGP attributes in vitro, including indole-3-acetic acid (IAA) production, the solubilization of phosphorus, biological nitrogen fixation, siderophores, and extracellular enzymatic activities (protease, lipase, chitin, and cellulase). Screening was carried out using the standard method (Li et al.; Mohamad et al.) [44,45]. The assays were performed with four repetitions of each strain. 

#### 2.1.1. Indole-3-Acetic Acid (IAA) Production Assay

Bacterial indole-3-acetic acid (IAA) production was examined by using the Salkowski reagent, as described by Li et al. [44]. Briefly, each strain was grown in 15 mL tryptophan broth and incubated at 28–30 °C in the dark for approximately five days at 130 rpm. Auxin production was determined by hydrolyzing the tryptophan into indole, which later on reacts with the Salkowski reagent to form a pink color. This mixture was incubated at room temperature for 30 min in the dark. The phytohormone was confirmed by measuring absorbance at OD_530_ nm by using a 96-well microplate reader and compared with known amounts of IAA using the Salkowski reagent and sterile tryptophan broth without bacterial inoculation as blanks [46,47].

#### 2.1.2. Phosphorus Solubilization Assay

The quantitative estimation of the ability of the bacterial endophytes to solubilize inorganic phosphate was carried out by using Pikovskaya’s agar media supplemented with 5 g precipitated tricalcium phosphate Ca_3_(PO_4_)_2_ and 0.025 g Bromophenol Blue per liter as an indicator, pH 7.0, according to the methodology described by da Cunha Ferreira et al.; Dubey et al. [46,48]. The endophytic bacteria that produced translucent halos around the bacterial colonies due to the utilization of tricalcium phosphate were considered positives for phosphate solubilizing [45,49].

#### 2.1.3. Biological Nitrogen Fixation

The ability of putative endophytic bacterial strains for nitrogen-fixing attributes was assessed through the inoculation of endophytic bacteria on two nitrogen-free media: Ashby’s mannitol agar and NFC medium [45]. Freshly grown cultures were inoculated separately on selective nitrogen-free media at 28 °C for 7 days. The nitrogenase activity was observed based on the colony growth on selective nitrogen-free media Ashby’s and NFC agar plates.

#### 2.1.4. Production of Siderophores by Endophytic Bacteria

To evaluate siderophore production, CAS medium (blue agar chrome azurol S (CAS), medium containing chrome azurol S (CAS), and hexadecyltrim ethylammonium bromide (HDTMA)) were used as described by Li et al. [44]. All tested strains (55) were inoculated into the CAS medium and incubated at 28 °C for 7 days. A change in media color from blue to a red/purple or orange/purple halo zone around the bacterial colony was scored as positive for the production of siderophores [44,45].

#### 2.1.5. Screening the Extracellular Enzymatic Activities 

The activity of extracellular enzymes (cellulase, protease, lipase, and chitinase) of 55 strains was assessed using the spot dot inoculation technique on selective agar media supplemented with specific substrates, depending on the enzyme being tested. Assays were performed with four repetitions of each strain, and the bacterial colonies that produced a halo zone around the colonies were considered positives [47]. The diameters of the halo zones and bacterial colonies were measured using the formula proposed by da Cunha Ferreira et al. [46]. Control treatments consisted of the same media without bacterial inoculation.

Chitinolytic activity was assessed by growing the endophytic bacterial strains on a colloidal chitin medium made from crab shells (C_8_H_13_NO_5_)**_n_** provided by Solarbio Life Science and following the protocol of Agrawal and Kotasthane.; Mohamad et al. [50,51]. Then, agar plates were incubated for 7 days at 28 °C. The appearance of clear/halo zones was observed around bacterial colonies, indicating the hydrolysis of chitin [45]. 

Cellulolytic activity was determined by spot-inoculating 24 h fresh bacterial cultures on CMC (carboxymethyl cellulose) agar plates and incubating them at 28 °C for 7 days. After incubation, plates were then flooded with a Congo red solution and then destained using an NaCl solution as an indicator [45,46]. The appearance of a clear zone around the bacterial colonies indicated cellulase hydrolysis activity [46].

The proteolytic activity of endophytic strains was assessed through inoculating bacterial cultures on skim milk agar 5% (*v*/*v*) medium. The skim milk agar plates were incubated for 5 days at 28 °C. The hydrolysis of skim milk was observed as a clear zone around the bacterial colonies, indicating protease activity [45].

The efficacy of endophytic bacteria microbes to produce lipase enzyme was investigated after the inoculation of bacterial cultures on modified Sierra lipolysis agar supplemented with 0.2 g ferrous citrate C_6_H_5_FeO_7_ and 3 g beef extract per liter for 7 days at 28 °C [45]. After the incubation period, the appearance of white calcium precipitates around the bacterial colonies indicated a positive reaction [45]. 

#### 2.1.6. Screening for Antagonistic Activity

The bacterial endophytes property of antagonism was evaluated against two widely prevailing phytopathogens represented as *F. oxysporum* and *V. dahlia* by using the dual plate culture method in vitro according to Mohamad et al.; Mahgoub et al. [51,52]. For bioassay, a 5 mm plug from the leading edge of a 5-day-old culture of each phytopathogen on PDA was placed in the center of the agar plate. After 24 h of incubation at 26 °C, each bacterial strain was spotted at 4 equidistant points along the perimeter of the PDA plate (3 plates per isolate) following procedures described by [46]. PDA plates containing only a mycelium disc without bacteria were used as control. Plates were incubated at 26 °C for 10 days. The antagonistic activity of bacterial isolates was evaluated by measuring the radial growth inhibition of phytopathogenic fungi. Relative inhibition (RI) was calculated using the formula below [51]: $$\mathrm{Inhibition}\;(\%)=\frac{F_{c}-T_{b}}{F_{c}-F_{0}}\times 100$$
where (F_c_) is the diameter of the fungal colony on the control, T_d_ is the diameter of the fungal colony on the PDA treatment, and (F_0_) is the diameter of test fungus agar discs (approximately 5 mm). 

### 2.2. Effect of the Most Potent Endophytic Bacteria on Plant Growth Parameters

#### 2.2.1. Greenhouse Experimental Design

To assess the impact of the cultivated beneficial endophytic actinobacterial strains associated with wild *T. roseus* in enhancing tomato and cotton growth parameters in low nutrient soil mixtures, a completely randomized pot experiment with three biological replicates per treatment and four seedlings per pot for each treatment was carried out to test the efficacy of the 10 most potent symbiotic endophytic bacteria on plant growth promotion properties. Those ten strains were positive for at least seven plant-beneficial traits in vitro, including nitrogenase activity, IAA production, phosphate solubilization, siderophores, and the production of at least two extracellular enzymes; therefore, they were selected for their better activities to test the effect of their inoculation on the growth performance of tomato and cotton as model plants [44,45]. Parallel controls were maintained by cultivating cotton and tomato with sterilized compost, watered with just water without bacterial inoculation, and irrigated with tap water whenever needed.

#### 2.2.2. Bacterial Inoculations and Soil Condition

Tomato seeds (*Solanum lycopersicum*. cv. Fuji Pink) and cotton (*Gossypium hirsutum* “Yumian-1”) were superficially disinfected by soaking in 70% ethyl alcohol for 5 min, followed by 2% sodium hypochlorite for 3 min, followed by five times of washing in sterile distilled water [45]. The sterilized seeds of tomato and cotton were germinated on wet filter papers and placed in sterilized glass Petri dishes (9 cm diameter). The Petri dishes were covered with parafilm tape to prevent evaporation and kept in the plant growth chamber at 25 °C for 4–5 days. 

For the preparation of bacterial suspensions, the ten most potent symbiotic endophytic bacteria were cultured individually in ISP_2_ broth at 28 °C, under constant agitation of 130 rpm for 4 days. Bacterial cultures were centrifuged, and cells were suspended in sterilized distal water and adjusted by using Densicheck plus (Biomerieux, Rodolphe, Durham, NC, USA) to a final concentration of 10^8^ CFU/mL [45]. 

Pre-germinated and surface-disinfected tomato and cotton seeds were submerged in respective bacterial suspensions at room temperature for 5 min with manual gentle shaking. After the microbiolization process, the seedlings were planted in pots filled with sterilized soil mixture—sand:perlite:compost:peat (1:1:1:1, *v*/*v*/*v*/*v*)—and the seedlings were grown in a greenhouse at 25–30 °C for 65 days and irrigated with tap water as required without adding any fertilizers [46]. After one week of germination, 10 mL of each bacterial suspension at 10^8^ CFU/mL was individually applied near the root zone of each plant.

#### 2.2.3. Estimation of Plant Growth Parameters

After the cotton and tomato were cultivated for 65 days, the plant growth-related parameters were measured, including the length of the aerial part, the length of the roots, the fresh mass of the roots, and the fresh mass of the aerial part. For dry mass determination, the stem and root were separated, followed by incubation in an oven at 65 °C for 24 h. Then, each part of the plant was weighed on an analytical balance.

### 2.3. Extraction and Identification of Metabolites

#### 2.3.1. Isolation and Purification of Bioactive Compounds

The antibiosis experiment was carried out by the co-cultivation of strain *Saccharopolyspora gregorii* (XIEG05), *Streptomyces enissocaesilis* (XIEG07), *Streptomyces enissocaesilis* (XIEG45), and *Streptomyces luteus* (XIEG51) with *V. dahliae* in 500 m/L of a liquid medium at 28 °C for 3 weeks with an agitation speed of 180 rpm in triplicate. Microbial cells were collected by centrifugation at 6000× *g* for 15 min. The supernatant was mixed with an equal volume (1:1) of ethyl acetate by vigorous shaking for 60 min and allowed to settle. These processes were repeated 3 times. Afterward, the organic solvent phase was evaporated at 43 ± 2 °C under a vacuum, using a rotary evaporator (IKA, HB10 basic). The ethyl acetate extract was dissolved in 5 mL of Tris-Cl buffer (0.02 M, pH 7.0) and used for gas-chromatography/mass spectrometry (GC-MS) [51].

#### 2.3.2. Identification of Bioactive Compounds

The GC-MS analysis of the endophytic actinobacteria cell-free extracts was performed using a gas chromatograph (Model 7890A, Agilent, Palo Alto, CA, USA) equipped with a split-splitless injector, an Agilent model 7693 autosampler, and an Agilent HP-5MS fused silica column (5% Phenyl-methylpolysiloxane, 30 m length, 0.25 mm I.D., film thickness 0.25 mm). The injecting volume of each sample was 1 µL, and the GC conditions included programmed heating from 50 to 300 °C °C/min, followed by 10 min at 300 °C. The injector was maintained at 280 °C. The carrier gas for GC was helium, at 1.0 mL min^−1^, and the split mode was 5:1. The GC was fitted with a quadrupole mass spectrometer with an Agilent model 5975 detector. The MS conditions were as follows: ionization energy, 70 eV; electronic impact ion source temperature, 230 °C; quadrupole temperature, 150 °C; scan rate, 3.2 scans/s; mass range, 50–1000 u. The compounds were identified based on the match with their mass spectra and retention indices with the NIST/Wiley 275 library (Wiley, New York, NY, USA). The relative abundance of each feature was calculated from the Total Ion Chromatogram (TIC) computationally [51]. 

### 2.4. Statistical Analysis

Data represent the mean of 10–12 replicates ± standard error (SE) calculated by Excel. One-way ANOVA was used to compare the means of root length (RL), root fresh weight (RFW), shoot length (SL), and shoot fresh weight (SFW) separately, and Tukey’s HSD post hoc test was used for multiple comparisons at α = 0.05. Statistical analyses were conducted in R (R Core Team (2022). R: A language and environment for statistical computing. R Foundation for Statistical Computing, Vienna, Austria. URL https://www.R-project.org/).

## 3. Results

### 3.1. Plant Growth-Promoting (PGP) Parameters

#### 3.1.1. Evaluation of PGP Traits

To further characterize the endophytic actinobacterial strains to select the most potent strains with multiple PGP as well as biocontrol traits, the PGP indicators were ammonia production, phosphate solubilization, siderophore production, the hydrolyzation of plant cells, and IAA production, which were all beneficial indicators for plant growth. The isolates were screened in vitro for plant growth-promoting traits (Appendix A). The PGP traits of the tested endophytic actinobacteria are summarized in (Figure 1).

Our results showed that the highest percentage (96.25%) of the endophytic actinobacterial strains were able to grow on N_2_ free medium. Among the 55 bacterial strains, the majority of positive strains, 32%, belong to the genera *Nocardiopsis*, followed by *Streptomyces* (28%), *Kocuria* (10%), and *Saccharomonospora* (10%) (Figure 1). 

Regarding siderophore production, the results have shown that 94.34% were able to produce a yellow-orange halo in CAS agar plates. Out of the 55 bacterial strains, the majority of positive strains, 30%, belong to the genera *Nocardiopsis*, followed by *Streptomyces* (28%), *Kocuria* (10%), and *Saccharomonospora* (10%) (Figure 1).

For phosphate solubilization, the results have shown that a lower number of isolates (24%) were able to solubilize phosphate. The highest percentage among the 55 endophytic actinobacterial strains able to solubilize phosphate belong to the genera *Streptomyces* (18%), followed by *Saccharomonospora* (4%), and *Micromonospora* (2%) (Figure 1).

The results of the experiment found that 47 (88.68%) endophytes could produce IAA. Out of the 55 endophytic bacteria, the most positive strains 30% belong to the genera *Nocardiopsis*, followed by *Streptomyces* (26%), *Kocuria* (10%), and *Saccharomonospora* (8%) (Figure 1). 

The investigation of the hydrolytic enzyme-producing ability of the tested strains indicated that protease, cellulase, and lipase producers were more prevalent among all the isolates (75.47, 81.13, 81.00%), respectively, and most of these strains belong to the genera *Nocardiopsis*, followed by *Streptomyces*, *Saccharomonospora*, and *Kocuria*. In addition, out of the total tested endophytic actinobacterial isolates, 18.87% were able to produce chitinase enzymes, and the majority of these strains belonged to the genera *Streptomyces* (16%), followed by *Saccharomonospora* (2%) (Figure 1).

#### 3.1.2. In-Vitro Antagonistic Bioassay

All endophytic actinobacterial strains (55) were individually tested in vitro against two fungal pathogens, *F. oxysporum* and *V. dahliae* (Figure 2). Our results showed that 45% of tested strains have an inhibitory effect on the phytopathogen *F. oxysporum* by inhibiting fungal growth in agar plate followed by 26% for *V. dahlia*. In this study, biocontrol activity among the positive isolates against the two fungal pathogens was only seen for 12 strains (Appendix A). The greatest inhibition percentages of fungal colony growth were noticed with a completely antagonistic nature against the two fungal pathogens were achieved due to treatment with the following bacterial endophytes: *Saccharopolyspora gregorii* (XIEG05), *Streptomyces enissocaesilis* (XIEG07), *Nocardiopsis dassonvillei* (XIEG45), and *Streptomyces luteus* (XIEG51) against the two fungal pathogens (Appendix A).

### 3.2. In-Planta Assay for Plant Growth Promotion Parameters by Selected Actinobacterial Strains

To verify the ability of bacterial endophytes from the medicine plant *T. vulgaris* for enhanced crop productivity and agriculture sustainability, we further analyzed the efficacy of potent endophytic actinobacterial strains for plant growth-promoting (PGP) in treated plants compared to the untreated control to investigate whether the growth promotion properties of the tested strains were restricted to cotton and tomato or not. In this pilot study, a total of ten strains (XIEG05, XIEG07, XIEG10, XIEG34, XIEG40, XIEG41, XIEG45, XIEG50, XIEG55, and XIEG63) belonging to two genera, *Streptomyces* and *Saccharopolyspora*, and positive for at least seven plant-beneficial traits in vitro, including nitrogenase activity, IAA production, phosphate solubilization, siderophores, and the production of at least two cell wall hydrolytic enzymes were applied to in planta assays to demonstrate potential plant growth stimulation properties in pot experiments with tomato and cotton plants (Appendix A). 

In the present study, most of the tested endophytic actinobacteria showed significant differences (*p* ≤ 0.05) in terms of plant growth-promoting (PGP) parameters in the treated plants regarding the length of the aerial part and fresh mass compared to the uninoculated plants. In addition, the results revealed some considerable differences regarding the effect of each bacterial strain on plant growth-related parameters. The degree of plant growth parameters in both tested plants varied drastically with various endophytic actinobacteria treatments and showed variance between the treated plants and the control. The growth parameters, including root and shoot length and root and shoot fresh and dry weight, were significantly increased in the treated tomato (Figure 3) and cotton (Figure 4). 

The maximum significant increase (*p* ≤ 0.05) was observed in the root lengths (RL) of the tomato inoculated with strain XIEG63 by 31.45% followed by strain XIEG55 by 17.95% (Figure 3A), while cotton showed an increase of 112.65% followed by strain XIEG55 by 95.78%, respectively, compared to the uninoculated plants (Figure 4A). On the contrary, among the 10 bacterial strains applied to tomato plants, three strains (XIEG10, XIEG45, and XIEG50) did not show any effects on the growth of tomato with respect to the control (Figure 3A).

Regarding root fresh weight (RFW) and root dry weight (RDW), a significant (*p* ≤ 0.05) increase in RFW and RDW was observed in tomato plants treated with strain XIEG63 by 82.22 and 76% followed by strain XIEG55 by 40.56 and 42.50%, respectively (Figure 3B,C), while cotton showed an increase of 213.53, 97%, followed by strain XIEG55 by 163.63, 82.50%, respectively, compared to the uninoculated plants (Figure 4B,C).

In these assays, the inoculation of the tomato plants with strain XIEG55 caused a great improvement in shoot length (SL) (*p* ≤ 0.05) by 58.08% followed by strain XIEG63 by 44.46% (Figure 3D), whereas for cotton, strain XIEG63 showed an increase of 88.15% followed by strain XIEG55 by 46.44%, compared to the uninoculated plants (Figure 4D).

All tested endophytic strains presented a significantly higher shoot fresh weight (SFW) and shoot dry weight (SDW) of both tested plants except strain XIEG10 for tomato, which did not show any effects on SFW and SDW with respect to the control (Figure 3E,F). In addition, the results revealed that strain XIEG55 demonstrated a significantly enhanced tomato shoot development (*p* ≤ 0.05) in SFW by 130.67% and SDW by 173.40%, followed by strain XIEG63 by 112.24 and 140.78%, respectively (Figure 3E,F), while for cotton, strain XIEG63 showed a maximum increase of 147.88% (SFW) and 110.23% (SDW) followed by strain XIEG41 by 140.52, 108.86%, respectively, compared to the uninoculated plants (Figure 4E,F).

Overall, for both crops, the data analysis in Figure 3 and Figure 4 shows that, in general, most of the tested endophytic strains after 60 days of the pot experiment promoted the growth of tomato and cotton by causing a higher increase in biomass in the aerial part and the roots, compared to the uninoculated plants. In particular, comparing the physiological parameters results of two plants inoculated with the same strains and under the same conditions revealed that the inoculation with strains XIEG40, XIEG55, and XIEG63 showed the best performance by enhancing the root length and biomass of tomato and cotton compared to the uninoculated plants. In addition, interestingly, the results revealed that the same strain XIEG63 was more efficient in promoting the aerial part than in the roots in tomato, but in contrast with cotton, compared with the control. However, the percentage of the increase in the cotton root length was higher than tomato. In particular, strain XIEG63 increases RL by 112%, while for tomato 31% and RFW by 213%, while for cotton, 82%.

### 3.3. Detection of Bioactive Compounds by GC-MS Analysis

The ethyl acetate extracts of four co-culture endophytic actinobacterial strains, Saccharopolyspora gregorii (XIEG05), *Streptomyces enissocaesilis* (XIEG07), *Streptomyces enissocaesilis* (XIEG45), and *Streptomyces luteus* (XIEG51) with *V. dahliae* were studied by GC-MS (Figure 5). Each peak represents an individual chemical compound, and the interpretation of mass spectrum GC-MS was conducted using the database of the National Institute Standard and Technology (NIST). 

The GC-MS analysis of strain *Saccharopolyspora gregorii* (XIEG05) showed 51 peaks (Figure 5A). The mass spectrum of strain XIEG05 showed that there were seven major compounds in ethyl acetate extracts suggestive of dibutyl phthalate, cyclotetracosane, 9-Tricosene, (Z)-, oleyl alcohol, trifluoroacetate, pyridine-3-carboxamide, oxime, *N*-(2-trifluoromethylphenyl)-, decahydro-8a-ethyl-1,1,4a,6-tetramethylnaphthalene, and octasiloxane, 1,1,3,3,5,5,7,7,9,9,11,11,13,13,15,15-hexadecamethyl, identified as the major peak from this fraction (Appendix A). 

For strain *Streptomyces enissocaesilis* (XIEG07), the GC-MS results showed 94 compounds in total (Figure 5B). Five of them were characterized by mass analyzer detector GC/MS as high peaks including phenylethyl alcohol, trans-1,10-dimethyl-trans-9-decalol, dibutyl phthalate, oleyl alcohol, trifluoroacetate, and heptasiloxane, 1,1,3,3,5,5,7,7,9,9,11,11,13,13-tetradecamethyl- (Appendix A). 

About 24 compounds were identified by GC-MS for strain *Streptomyces enissocaesilis* (XIEG45) (Figure 5C). The mass spectrum showed that five major peaks were obtained from ethyl acetate extracts suggestive of maltol, phenylethyl alcohol, benzeneacetic acid, pyrrolo [1,2-a] pyrazine-1,4-dione, hexahydro-3-(2-methylpropyl)-, and dibutyl phthalate (Appendix A).

For strain *Streptomyces luteus* (XIEG51), the GC-MS identified 46 compounds in total (Figure 5D). Eight of them were characterized as high peaks including maltol, phenylethyl alcohol, benzeneacetic acid, benzeneethanol, 4-hydroxy-, pyrrolo [1,2-a] pyrazine-1,4-dione, hexahydro-3-(2-methylpropyl)-, dibutyl phthalate, nonadecyl trifluoroacetate, and 1-docosene (Appendix A). Furthermore, several minor peaks were detected in the ethyl acetate extracts of four strains XIEG05, XIEG07, XIEG45, and XIEG51, as presented in Appendix A. Our results show that most of the compounds revealed by GC-MS were mainly phenols, fatty acid esters, and aromatic chemicals and have been reported to have antibacterial and antifungal activity.

## 4. Discussion

Global food production systems, especially the agriculture sector, are predicted to be adversely affected by climate change and the increasing human population. At the same time, attention has been focused on environmentally friendly (fewer chemical fertilizers) and ecologically sustainable management practices in the agriculture sector [13]. Endophytes can play important beneficiary roles in plant development directly, indirectly, or synergistically [48]. Therefore, the further manipulation of bacterial endophyte–plant interactions opens up a new and alternative sustainable strategy for managing agricultural practices, stimulating plant growth, and helping to limit the use of chemical fertilizers in the farming system [8].

In the present study, we obtained 55 actinobacterial strains associated with wild *T. roseus* from the Ili site representing 2 classes, 8 orders, 12 families, 18 genera, and 37 species [43] to evaluate the efficacy of these endophytic bacteria for enhancing plant growth and the antagonistic activity of phytopathogenic fungi in vitro. The findings from our study demonstrate the potential involvement of culturable endophytes of *T. roseus* in a broad range of biological processes with potential applications including plant growth stimulation, defense mechanisms, and the production of secondary metabolites. Here in this study, most of our endophytic isolates were able to possess direct PGP traits, including auxin (88.68%), ammonia (96.25%), siderophore production (94.34%), and phosphate solubilization (24%), along with cell-wall degrading activities such as protease (75.47%), cellulase (81.13%), lipase (81.00%), and chitinase (18.87%). Our results are in concordance with the studies conducted by [28,45,46,49,53,54].

On the other hand, in vitro screens for antagonistic activity were conducted by co-cultivating the *T. roseus* endophytes with common fungal pathogens of tomato (*F. oxysporum* f. sp) and cotton (*V. dahliae*). In these assays, 45% of tested strains have an inhibitory effect on the phytopathogen *F. oxysporum* by inhibiting fungal growth in agar plate followed by 26% for *V. dahlia*. These results are in agreement with the previous findings of [52,54,55].

One of the major objectives of this pilot study was to investigate whether the growth promotion properties of the tested strains were restricted to cotton and tomato or not. To test this hypothesis, a total of ten endophytic strains belonging to two genera, *Streptomyces* and *Saccharopolyspora*, and positive for at least seven plant-beneficial traits in vitro were applied to *in planta* assays to demonstrate potential plant growth stimulation properties in a controlled soil–plant system in pot experiments with tomato and cotton to select the most potent strain with the best performance for both tested plants. In the present investigation, changes in the length of the aerial part and fresh mass were visually observed in tomato and cotton plants inoculated with bacterial endophytes in comparison to control plants. The maximum significant increase (*p* ≤ 0.05) was observed in the aerial part and fresh mass of the cotton inoculated with strain XIEG63 belonging to *Saccharopolyspora taberi*, compared to the uninoculated plants (Figure 3). On the other hand, Interestingly, for tomato, strain XIEG63 showed a maximum significant increase (*p* ≤ 0.05) in root length (RL), root fresh weight (RFW), and root dry weight (RDW), while strain XIEG55, belonging to *Streptomyces viridochromogenes*, showed the highest increase in shoot length (SL), shoot fresh weight (SFW), and shoot dry weight (SDW) compared to the uninoculated plants (Figure 4). The data indicated that the strains XIEG63 and XIEG55 were able to possess direct PGP traits including auxin, ammonia, siderophore production, and phosphate solubilization along with cell-wall degrading activities such as protease, cellulase, lipase, and chitinase (Appendix A). Thus, strains XIEG63 and XIEG55 were the potent strains with various plant growth-promoting traits that promoted plant heights and the fresh and dry weight of shoots and roots compared to uninoculated plants (control), which means that these two strains can be used as potent plant growth promoters due to their ability to enhance the plant growth and development by increasing the availability of key nutrients such as ammonia, iron, and phosphate solubilization, which are important for plant growth [16]. In addition, producing phytohormones by bacterial endophytes such as IAA stimulates root elongation and induces the formation of adventitious roots [56]. Similar observations have been reported in previous studies, where plant growth parameters were enhanced by co-inoculation with endophytes, which can be an important strategy for enhancing crop production in a more eco-friendly and cost-effective mode in many regions around the world, for crops such as wheat [57], lettuce [58], canola [59], peanuts [60], cowpea [46], maize [61], sunflower [62], tea [15], and rice [63].

A key first step in the identification of novel biocontrol agents is the screening of antagonistic activities. Our data showed that the greatest inhibition percentages of fungal colony growth against the two fungal pathogens were achieved due to treatment with strains XIEG05, XIEG07, XIEG45, and XIEG51, which clearly shows that endophytic actinobacteria associated with the medicinal plant *T. roseus* can have a greater potential of antagonistic activities. Our results partially support the hypothesis that the medicinal properties of herbal medicine could be due to the existence of beneficial endophytes in the host [64]. This antagonistic activity of endophytes could be related to different ecological mechanisms, such as producing secondary metabolites that inhibit fungal growth and development, or competition between endophytes and pathogens around nutrients [65]. These findings are in agreement with previous reports where endophytic actinobacteria from different medicinal plants are reported as a major source of natural products with potential antifungal activity [27,66].

In this investigation, among all antagonistic endophyte strains, XIEG05, XIEG07, XIEG45, and XIEG51 were selected for exometabolomic studies by GC-MS (Figure 5) based on their greatest ability to inhibit fungal colony growth. The GC-MS analysis of crude extracts buffered from *Saccharopolyspora gregorii* strain (XIEG05) showed six major compounds in ethyl acetate extracts, suggestive of antifungal compounds such as cyclotetracosane, 9-tricosene, (Z)-, and dibutyl phthalate [67], and antimicrobial compounds such as pyridine-3-carboxamide [68], trifluoroacetate [69], oxime, *N*-(2-trifluoromethylphenyl)- [70], and octasiloxane,1,3,3,5,5,7,7,9,9,11,11,13,13,15,15-hexadecamethyl- [71] (Appendix A).

For *Streptomyces enissocaesilis* (XIEG07), the GC-MS showed a total of 94 compounds (Figure 5B). Five of them were characterized by mass analyzer detector GC-MS as high peaks including antimicrobial compounds such as phenylethyl alcohol [72] and trifluoroacetate [69]; antifungal compounds such as dibutyl phthalate [67], heptasiloxane, and 1,1,3,3,5,5,7,7,9,9,11,11,13,13-tetradecamethyl- [73]; and anti-tumor agent compound oleyl alcohol [74] (Appendix A).

About 24 compounds (Figure 5C) were identified by mass spectrum for *Streptomyces enissocaesilis* (XIEG45). Among them, five major peaks were obtained from ethyl acetate extracts suggestive of antimicrobial agent compounds such as phenylethyl alcohol [72] and benzeneacetic acid [75]; antifungal compounds such as pyrrolo [1,2-a] pyrazine-1,4-dione [76] and hexahydro-3-(2-methylpropyl)- [77]; and dibutyl phthalate [67] (Appendix A).

For *Streptomyces luteus* (XIEG51) the GC-MS showed a total of 46 compounds (Figure 5D). Eight of them were characterized by mass analyzer detector GC-MS as major peaks including antifungal compounds such as phenylethyl alcohol [72], pyrrolo [1,2-a] pyrazine-1,4-dione, and hexahydro-3-(2-methylpropyl)- [76] and antimicrobial agent compounds such as benzeneacetic acid [75], benzeneethanol, 4-hydroxy [78], dibutyl phthalate [67], nonadecyl trifluoroacetate, and 1-docosene [79,80] (Appendix A).

## 5. Conclusions

Endophytes can play important beneficiary roles in plant development, directly, indirectly, or synergistically. Here in this study, the majority of our endophytic isolates were able to possess direct PGP traits, including auxin (88%), ammonia (96%), siderophore production (94%), and phosphate solubilization (24%), along with cell-wall degrading activities such as protease (75%), cellulase (81%), lipase (81%), and chitinase (18%). Most positive strains for multiple beneficial traits belonged to the genera *Nocardiopsis*, followed by *Streptomyces*. On the other hand, in vitro screens for antagonistic activity were conducted by co-cultivating the *T. roseus* endophytes with common fungal pathogens of tomato (*F. oxysporum*) and cotton (*V. dahliae*). In these assays, 45% of tested strains had an inhibitory effect on the phytopathogen *F. oxysporum* by inhibiting fungal growth in agar plate followed by 26% for *V. dahlia*. Overall, in both crops, data analysis showed that, in general, most of the tested endophytic strains after 60 days of the pot experiment promoted the growth of tomato and cotton plants by causing a higher increase in biomass in the aerial part and the roots, compared to the uninoculated plants. In particular, comparing the physiological parameters results of two plants inoculated with the same strains and under the same conditions revealed that the inoculation with strains XIEG55 and XIEG63 showed the best performance by enhancing the root length and biomass of tomato and cotton, compared to the uninoculated plants. A key first step in the identification of novel biocontrol agents is the screening of antagonistic activities. Our data showed that the greatest inhibition percentages of fungal colony growth were noticed with a completely antagonistic nature against all the two fungal pathogens were achieved due to treatment with strains XIEG05, XIEG07, XIEG45, and XIEG51, which clearly shows that endophytic actinobacteria associated with medicinal plants can have a greater degree of antagonistic activities. The GC-MS analysis showed that most of the compounds revealed by GC-MS were mainly fatty acid esters, phenols, alkanes, alkenes, and aromatic chemicals and have been reported to have antifungal activity. For example, strain XIEG07 resolved 94 compounds in total; three of them were characterized by mass analyzer detector GC/MS as high peaks including antifungal compounds such as dibutyl phthalate, heptasiloxane, and 1,1,3,3,5,5,7,7,9,9,11,11,13,13-tetradecamethyl-. Our investigation emphasizes that endophytic actinobacteria associated with medicinal plants might help reduce the use of chemical fertilization and can be used as environmentally friendly bio-inoculants. In addition to these emerging approaches, if issues linked to policy development and the social acceptability of microbial products can be simultaneously addressed, especially for smallholder agroecosystems, these bio-based tools can potentially contribute significantly to enhanced crop productivity and agriculture sustainability

## Figures and Tables

**Figure 1 microorganisms-10-01802-f001:**
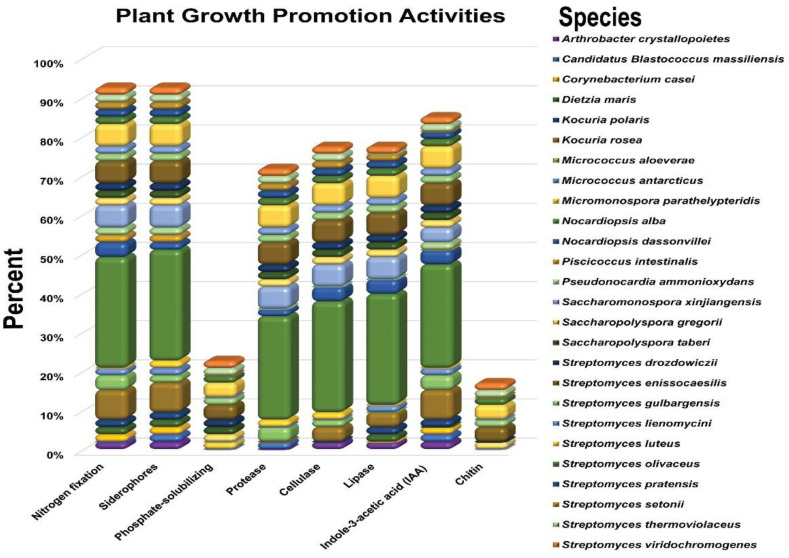
Plant growth-promotion traits of endophytic actinobacteria species from herbal medicinal plant *Thymus roseus* in vitro. All experiments were performed twice with three replicates for each individual strain.

**Figure 2 microorganisms-10-01802-f002:**
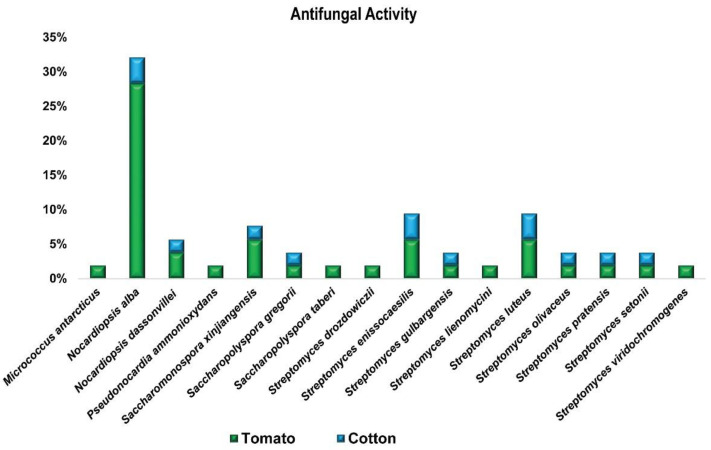
Antifungal activity of endophytes against two common fungal pathogens, F1: *F. oxysporum*, F2: *V. dahliae*. Each ring, F1 and F2, represents the total number of strains with antagonistic activity against the fungal pathogen and is divided into different colors based on the proportion of each species to that total.

**Figure 3 microorganisms-10-01802-f003:**
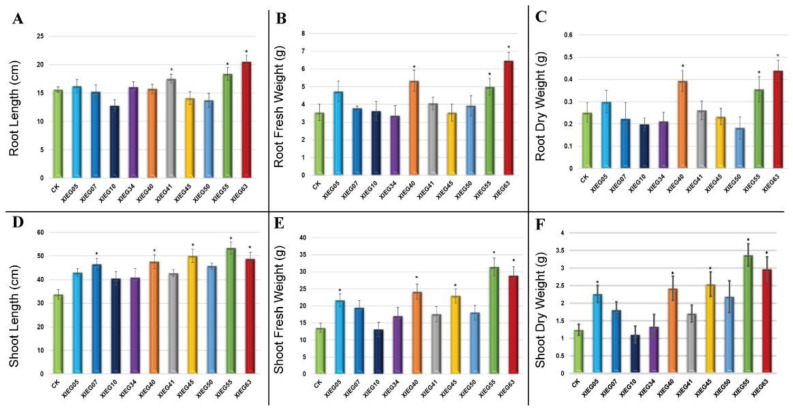
The growth parameters of tomato 60 days after inoculation with the selected endophytic actinobacteria compared with the control. The data represent a mean of 10–12 replicates ± standard error (SE). The columns marked by (“*”) indicate significant differences based on one-way ANOVA, followed by Tukey’s HSD post hoc test for multiple comparisons at alpha level = 0.05. (**A**) Root length; (**B**) Root fresh weight; (**C**) Root dry weight; (**D**) Shoot length; (**E**) Shoot fresh weight; (**F**) Shoot dry weight.

**Figure 4 microorganisms-10-01802-f004:**
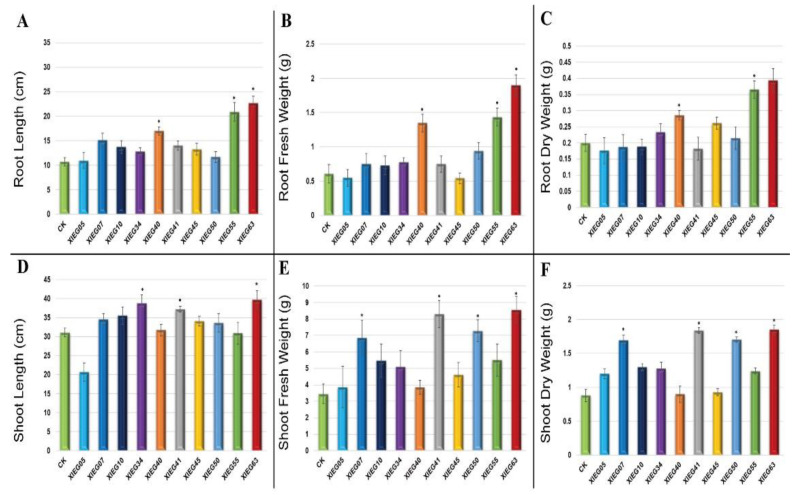
The growth parameters of cotton 60 days after inoculation with the selected endophytic actinobacteria compared with an uninoculated control plant. The data represent a mean of 10–12 replicates ± standard error (SE). The columns marked by (“*”) indicate significant differences based on one-way ANOVA, followed by Tukey’s HSD post hoc test for multiple comparisons at alpha level = 0.05. (**A**) Root length; (**B**) Root fresh weight; (**C**) Root dry weight; (**D**) Shoot length; (**E**) Shoot fresh weight; (**F**) Shoot dry weight.

**Figure 5 microorganisms-10-01802-f005:**
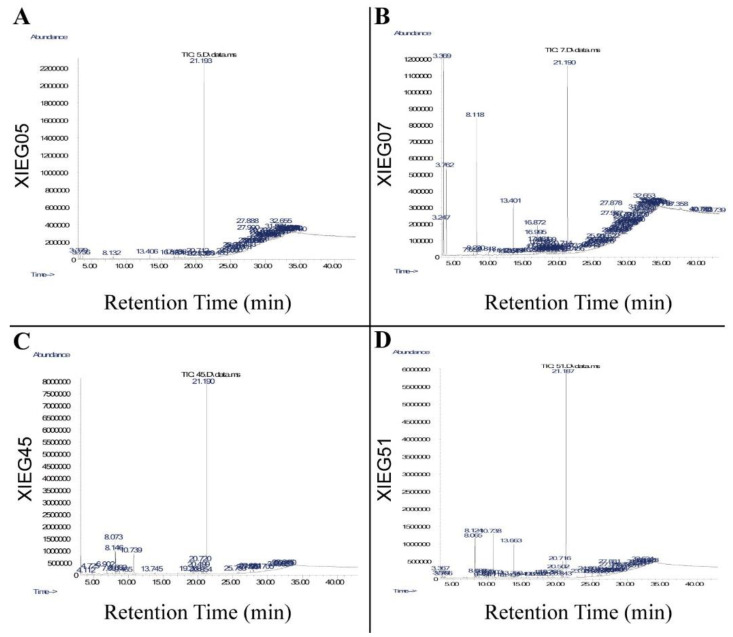
GC-MS analysis of potential bioactive compounds in ethyl acetate extracts of cell supernatant buffered of four co-cultures endophytic actinobacterial strains, (**A**) *Saccharopolyspora gregorii* (XIEG05), (**B**) *Streptomyces enissocaesilis* (XIEG07), (**C**) *Nocardiopsis dassonvillei* (XIEG45), and (**D**) *Streptomyces luteus* (XIEG51) with *V. dahliae*.

## Data Availability

Raw sequence data reported in this paper have been deposited in the GenBank in the NCBI under accession numbers (MN686608–MN686629(22), MN686648–686678(31), and MN688648–688649(2)).

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
