# Peer review of "The Metabolic Potential of Endophytic *Actinobacteria* Associated with Medicinal Plant *Thymus roseus* as a Plant-Growth Stimulator"

_microorganisms, 2022, doi:10.3390/microorganisms10091802_

Round 1

Reviewer 1 Report

The work entitled “The metabolic potential of endophytic actinobacteria associated with medicinal plant Thymus roseus as plant growth stimulator, authors are Osama Abdalla Abdelshafy Mohamad, Yonghong Liu, Yin Huang, Li Li, Jinbiao Ma, Dilfuza Egamberdieva, Lei Gao, Baozhu Fang, Shaimaa Hatab, Hongchen Jiang and Wenjun Li, is devoted to the actual problem of crop growth stimulation using biologically and ecologically safe methods. The 55 endophyte actinobacterial strains isolated from the medical plant Thymus roseus were screened for their plant growth stimulating and antifungal activities. The appropriate number of tests including determination of hydrolase activities, anatagonistic effects, estimation of plant growth stimulation using various parameters and identification of secondary exometabolites was done. Endophyte strains with high plant growth stimulating and antifungal activities were selected as promising components of future biofertilizers.

However, the text requires major revision, see specific comments below.

Major revisions

1) It is strongly recommended to check English properly, if it is possible with a native English speaker. Some sentences are wrong or not understandable (for example, lines 135-136, 174-177, 219-222, 297-299).

2) Latin names should be checked properly throughout the text. Genus names and species epithets must be italicized always. At first appearance in the text, genus name must be written in full, at further appearances – shortened until the first capital letter. For example, lines 129-130 and line 203 – Verticillium dahliae and Fusarium oxysporum must be written as V. dahliae and F. oxysporum since it is not their first appearance in the text.

3) It is recommended to check various names if they start from capital or small letter. For example, in line 87 – why Curcuma, not curcuma? Many GC-MS detected compound start from capital letters, what is the reason for this?

4) Lines 30, 199, 476, 559 – Do authors mean “cellulase”, not “cellulose”?

5) In Material and Methods, it is recommended to give more details about strain isolation and identification, at least, indicate species or genera of isolated endophytes. However, this information belongs to a previously done work (the reference is given) but details are required to not re-read the reference [43]. Moreover, it is interesting that so many (55) endophyte actinobacterial strains were isolated from one plant species T. roseus. Is it the evidence that actinobacteria dominate among T. roseus endophytes?

6) Lines 196-200 – Is it described test for lipase, or protease, activity?

7) Figures 1, 2 – It will be more informative if number of positive strains will be shown in segments of diagrams, and the total number of strains of each species will be shown in legends.

8) Figure 1 – It is recommended to sign the y axis (name of the value and its units) and decipher N2, CAS and IAA in the figure caption. As follow from the Figure 1, can authors state that PGP activities are species, not strain, specific?

9) Line 368 + Figures 3, 4 – It seems that “significantly increased” is not really correct. Many strains did not show plant stimulating effects, and for others this effect is not really significantly higher than in control.

10) Figure 5 – It is strongly recommended to show a GC-MS chromatogram for control and blank sample to confirm that detected compounds originate from exometabolites not from the dirty solvent, plastic/glassware or GC-MS column.

11) What are mechanisms of antifungal activities of GC-MS detected compounds?

12) Authors state that one of aims is to determine if isolated actinobacterial endophytes have PGP activities only towards tomato and cotton or towards other plants, but no other plants besides tomato and cotton were used in the study.

Minor revisions

1) Abstract:

line 25 – considers or consiers?

lines 31, 32 – here, it is recommended to write genus names in full;

line 32 – “both crops” – what crops authors mean? Specific crops were not indicated before this line.

2) Lines 84-85 – why (root length, shoot 84 length, fresh and dry root, and shoot biomass) are in parentheses?

3) Line 150 – only IAA can be left since it was abbreviated earlier.

4) Line 194 – “zone”, not “zoon”.

Summary

Reconsider after major revision.

Author Response

Thank you so much for reviewing our paper. We are very sorry for delaying our revision because our city lockdown due to the epidemic of COVID19 again, so we are not able to go to work.

Major revisions

1) It is strongly recommended to check English properly, if it is possible with a native English speaker. Some sentences are wrong or not understandable (for example, lines 135-136, 174-177, 219-222, 297-299).

Thank you very much for your comment, we have corrected all confusing sentences such as (lines 135-136, 174-177, 219-222, 297-299). In addition, a native English speaker goes through the manuscript and improved the language. Please kindly check our revised manuscript.

2) Latin names should be checked properly throughout the text. Genus names and species epithets must be italicized always. At first appearance in the text, genus name must be written in full, at further appearances – shortened until the first capital letter. For example, lines 129-130 and line 203 – Verticillium dahliae and Fusarium oxysporum must be written as V. dahliae and F. oxysporum since it is not their first appearance in the text.

Thank you very much for your comment. Yes, you are right. Sorry about this typing mistake. We have italicized all Latin names and corrected fungus names to V. dahliae and F. oxysporum. Please kindly check our revised manuscript.

3) It is recommended to check various names if they start from capital or small letter. For example, in line 87 – why Curcuma, not curcuma? Many GC-MS detected compound start from capital letters, what is the reason for this?

Thank you very much for your comment. Yes, you are right. We have corrected all GC compounds in the main text and not starting from a capital letter and also corrected Curcuma to curcuma as well. In fact, we think that compounds start from capital letters to be easy to read by audiences. Please kindly check our revised manuscript.

4) Lines 30, 199, 476, 559 – Do authors mean “cellulase”, not “cellulose”?

Thank you very much for your comment. Yes, you are right. Sorry about this typing mistake. We have corrected “cellulose” to “cellulase”. Please kindly check our revised manuscript.

5) In Material and Methods, it is recommended to give more details about strain isolation and identification, at least, indicate species or genera of isolated endophytes. However, this information belongs to a previously done work (the reference is given) but details are required to not re-read the reference [43]. Moreover, it is interesting that so many (55) endophyte actinobacterial strains were isolated from one plant species T. roseus. Is it the evidence that actinobacteria dominate among T. roseus endophytes?

Indeed! Thank you for pointing this out! sorry about that. The corrections have been made throughout the Material and Methods section. In addition, we already provided the accession numbers of 55 strains in the Data Availability Statement section.  Please kindly check the new version of our manuscript.

6) Lines 196-200 – Is it described test for lipase, or protease, activity?

Thank you very much for your comment. Yes, it's for lipase test. The corrections have been made in the revised manuscript by using MS track changes.  

7) Figures 1, 2 – It will be more informative if number of positive strains will be shown in segments of diagrams, and the total number of strains of each species will be shown in legends.

Thank you very much for your comment. However, figure one showed the results of multiple Plant growth-promotion traits of 55 endophytic actinobacterial strains. We have to change the figures to be more informative. Please kindly check the new figures.

8) Figure 1 – It is recommended to sign the y axis (name of the value and its units) and decipher N2, CAS and IAA in the figure caption. As follow from the Figure 1, can authors state that PGP activities are species, not strain, specific?

Thank you for pointing this out! sorry about that. We have deciphered N2, CAS and IAA in figure1 and wrote the full name. in addition, we corrected the figure caption to (Figure 1. Plant growth-promotion traits of endophytic actinobacterial species from herbal medicinal plant Thymus roseus in vitro. Please kindly check the new version of our manuscript.

9) Line 368 + Figures 3, 4 – It seems that “significantly increased” is not really correct. Many strains did not show plant stimulating effects, and for others this effect is not really significantly higher than in control.

Indeed! Thank you for pointing this out! sorry about that. Yes, we have checked the significance again and we found that some mistakes in figures and corrections have been made. The column marked by ("*") indicate significant differences between different strains and control. Please kindly check the new version of our manuscript.

10) Figure 5 – It is strongly recommended to show a GC-MS chromatogram for control and blank sample to confirm that detected compounds originate from exometabolites not from the dirty solvent, plastic/glassware or GC-MS column.  What are mechanisms of antifungal activities of GC-MS detected compounds?

Thank you very much for your comment. In our study, we used a new solvent and all glassware was cleaned with methanol and ethyl acetate and then dried in an oven before being used for extraction. Moreover, one of the main objectives of this study is to identify the major antifungal and antimicrobial compounds produced by endophytes by GC in order to explain the mechanisms of antifungal activities in vitro conditions.

12) Authors state that one of aims is to determine if isolated actinobacterial endophytes have PGP activities only towards tomato and cotton or towards other plants, but no other plants besides tomato and cotton were used in the study.

Thank you so much for your comment, we only studied 2 plants and we did not study another plant. the present research aimed to assess the impact of cultivated beneficial endophytic actinobacterial strains associated with wild T. roseus in enhancing tomato and cotton crop yield grown in low nutrient soil mixtures, with the aim of investigating whether the growth promotion properties of the tested strains were restricted to cotton and tomato or not. In addition, our results showed that strains XIEG63 and XIEG55 were the potent strains with various PGP traits that caused a higher increase significantly (p ≤ 0.05) of length and biomass in the aerial part and roots of tomato and cotton, compared to the un-inoculated plants.

Minor revisions

1) Abstract:

line 25 – considers or consiers?

Done

lines 31, 32 – here, it is recommended to write genus names in full;

Done

line 32 – “both crops” – what crops authors mean? Specific crops were not indicated before this line.

Thank you for pointing this out! sorry about that. We deleted “both crops” and we named the plants used in this study (Overall, our results showed that strains XIEG63 and XIEG55 were the potent strains with various PGP traits that caused a higher increase significantly (p ≤ 0.05) of length and biomass in the aerial part and roots of tomato and cotton, compared to the un-inoculated plants). Please kindly check the new version of our manuscript.

2) Lines 84-85 – why (root length, shoot 84 length, fresh and dry root, and shoot biomass) are in parentheses?

Done

3) Line 150 – only IAA can be left since it was abbreviated earlier.

Done

4) Line 194 – “zone”, not “zoon”.

Done

Reviewer 2 Report

The manuscript needs linguistic correction. Some sentences should be rewritten to make them easier and better understood.

The numbering of subsections in the materials and methods section is incorrect. Subsection 2.1 is presented twice, under two different names.

Line 144 - the name of the author should be mentioned

Lines 169, 180, 184, 204, 208 - the same comment

Line 236 - Was the last water checked for control of the sterilization process??

Line 262 - How were the strains assigned to those species?

Line 291-293 - it should be presented as reference (R Core Team (2018). R: A language and environment for statistical computing. R Foundation for Statistical Computing, Vienna, Austria. URL https://www.R-project.org/)

Figure 1 is not very readable. The chart should be changed to another form.

Figure 2 does not carry all the information needed. In my opinion, the results should be presented in a different form (for example, a bar chart for pathogens). We do not know how many representatives there were of the different taxa analyzed.

Author Response

Thank you so much for reviewing our paper. We are very sorry for delaying our revision because our city lockdown due to the epidemic of COVID19 again, so we are not able to go to work.

The manuscript needs linguistic correction. Some sentences should be rewritten to make them easier and better understood.

Thank you very much for your comment, we have corrected all confusing sentences (lines 135-136, 174-177, 219-222, 297-299). In addition, a native English speaker goes throughout the manuscript and improved the language. Please kindly check our revised manuscript.

The numbering of subsections in the materials and methods section is incorrect. Subsection 2.1 is presented twice, under two different names.

Thank you for pointing this out. Sorry about this mistake. We have corrected it and removed the duplicate subtitles. Please kindly check the new version of our manuscript.

Line 144 - the name of the author should be mentioned

Lines 169, 180, 184, 204, 208 - the same comment

Thank you very much for your comment. We have added the author’s name. Please kindly check the new version of our manuscript.

Line 236 - Was the last water checked for control of the sterilization process??

Thank you so much for your question. Yes, we checked the final rinse of surface-sterilization distilled water and No microbial growth was detected on the ISP2 media after 3 days of incubation at 28C. This result indicated that the surface sterilization protocol was successful. In addition, we sterilized the soil mixture at 121C for 1 hour.

Line 262 - How were the strains assigned to those species?

Thank you so much for your question. In Material and Methods, we mention that the strains investigated in this study were isolated and identified by us in previous work (All bacterial endophytes 55 strains included two classes, eight orders, 12 families, 18 genera, and 37 species isolated from Ili site [43] were screened for direct PGP attributes in vitro, including Indole-3-acetic acid (IAA) production, solubilization of phosphorus, biological nitrogen fixation, siderophores, and extracellular enzymatic activities (protease, lipase, chitin, and cellulase) were carried out using the standard method (Li et al; Mohamad et al.) [44,45]). In addition, we already provided the accession numbers of 55 strains in Data Availability Statement section. Please kindly check the new version of our manuscript.

Line 291-293 - it should be presented as reference (R Core Team (2018). R: A language and environment for statistical computing. R Foundation for Statistical Computing, Vienna, Austria. URL https://www.R-project.org/)

Thank you very much for your comment. We have corrected it as you kindly suggested. Please kindly check the new version of our manuscript.

Figure 1 is not very readable. The chart should be changed to another form.

Thank you very much for your comment. We improved the quality of the figure and also decipher N2, CAS, and IAA in figure1 and wrote the full name. In addition, we corrected the figure caption to (Figure 1. Plant growth-promotion traits of endophytic actinobacterial species from herbal medicinal plant Thymus roseus in vitro). Please kindly check the new version of our manuscript.

Figure 2 does not carry all the information needed. In my opinion, the results should be presented in a different form (for example, a bar chart for pathogens). We do not know how many representatives there were of the different taxa analyzed.

Thank you very much for your comment. We have changed the figure style to a bar chart as you kindly suggested. Please kindly check the new version of our manuscript.

Round 2

Reviewer 1 Report

All necessary corrections were made after revision, and paper can be accepted for publication in current form.